# Telepsychiatry to Provide Mental Health Support to Healthcare Professionals during the COVID-19 Crisis: A Cross-Sectional Survey among 321 Healthcare Professionals in France

**DOI:** 10.3390/ijerph181910146

**Published:** 2021-09-27

**Authors:** Clément Cormi, Stéphane Sanchez, Valentine de l’Estoile, Laura Ollivier, Aude Letty, Gilles Berrut, Emmanuel Mulin

**Affiliations:** 1Pôle Territorial Santé Publique et Performance des Hôpitaux Champagne Sud, Centre Hospitalier de Troyes, 10000 Troyes, France; stephane.sanchez@hcs-sante.fr; 2LIST3N/Tech-CICO, Troyes University of Technology, 10010 Troyes, France; 3Fondation Korian pour le Bien-Vieillir, 75008 Paris, France; valentine.delestoile@korian.fr (V.d.l.); laura.ollivier@korian.fr (L.O.); aude.letty@korian.com (A.L.); gilles.berrut@icloud.com (G.B.); 4Pôle Hospitalo-Universitaire de Gérontologie Clinique, CHU Nantes, 44000 Nantes, France; 5Département de Médecine Gériatrique, UFR Sciences et Techniques Médicales, Université de Nantes, 44000 Nantes, France; 6Clinique Korian le Val du Fenouillet, 83260 La Crau, France; emmanuel.mulin@korian.fr; 7Laboratoire IAPS, Université de Toulon, 83130 Toulon, France

**Keywords:** mental health, telemedicine, telepsychiatry, COVID-19, pandemics

## Abstract

Pandemics are difficult times for the mental health of healthcare professionals, who are more likely to present with PTSD-like symptoms. In the context of a highly contagious communicable disease, telemedicine is a useful alternative to usual care, and should be considered as a means to support healthcare professionals’ mental health. This is a multicenter (*n* = 19), cross-sectional study, based on a 27-item questionnaire, aiming to investigate the acceptability to healthcare workers of a telepsychiatry service as a means of providing mental health support during the COVID-19 pandemic. Between October and December 2020, 321 responses were received, showing that women, caregiving staff, and those directly involved in the care of COVID-19 patients are less favorable to the idea of receiving remote support. In our population, barriers were related to the clinical setting or ethics, and most of the respondents would not accept a drug prescription by telepsychiatry. Although telepsychiatry should be a part of the armamentarium of mental health management, it is not suitable as a stand-alone approach, and should be combined with conventional face-to-face consultations.

## 1. Introduction

A new coronavirus disease emerged in December 2019 in China [1] and by March 2020, the World Health Organization (WHO) had already qualified the 2019 coronavirus disease, namely COVID-19, as a pandemic [2]. By May 2021, there had been almost 153 million confirmed cases around the world, causing over 3.2 million deaths [3].

Since the 20th century, the question of the mental health of frontline healthcare workers (HCWs) during epidemics has garnered increasing attention. During the Spanish influenza pandemic of 1919, there were published descriptions of the physical consequences for HCWs involved in patient management [4], but few data about the impact of the pandemic on the mental health of HCWs. During the epidemic of severe acute respiratory syndrome (SARS) in 2003, some studies reported the prevalence of mental health symptoms and post-traumatic stress disorder (PTSD) among frontline HCWs to be between 17% and 25% [5,6]. The presence of these disorders seemed to be higher in zones delivering acute care to infected persons (or “hotspots”). The COVID-19 pandemic had not long been under way when Lai et al. first reported mental health findings from the Chinese province of Hubei, which was the epicentre of the epidemic at the time. They described a high prevalence of depression symptoms (50.4%), anxiety (44.6%), insomnia (34%), and distress (74.5%) among HCWs [7]. Rapidly, similar reports followed from other countries, in line with the spread of the disease [8,9,10,11]. In a review of the psychological impact of epidemic and pandemic outbreaks on the mental health of HCWs, Preti et al. reported that HCWs have suffered a greater burden of psychopathological or psychological symptoms during the COVID-19 outbreak than during the SARS and MERS outbreaks, especially for PTSD symptoms, depression, and anxiety [12]. Previous epidemics have taught us that non-work-related determinants also impact the mental health of HCWs (e.g., family, network, and community/society-level indicators) [13]. However, the work organization has also been shown to affect mental health. Indeed, a lack of organizational emotional support is associated with a higher risk of psychological morbidity and distress [14]. In this regard, the American Psychiatric Association has published a guide for actions and activities that healthcare organizations could undertake to support the well-being of their physician workforce during the pandemic and beyond. One of these recommendations is to “provide timely, easy, non-stigmatized access to emotional support and mental health care for all physicians” [15].

In the context of a highly contagious communicable disease, telemedicine is a useful alternative to usual care [16], with potential applications in epidemic situations [17]. The recommendations for social distancing and lockdown during the COVID-19 pandemic led to a surge in the use of telemedicine solutions [18].

Telepsychiatry is defined as “the delivery of healthcare and the exchange of healthcare information for the purposes of providing psychiatric services across distances” [19]. It reportedly offers an equivalent quality of care as compared to face-to-face consultations [20]. In a review of the literature, Norman described the potential effects of using telepsychiatry [21], both in terms of the pathologies that are amenable to its use (depression, schizophrenia, etc.) and the therapeutic options that can be proposed (cognitive/behavioural therapies, psychotherapy, etc.), with a high level of patient satisfaction.

During the COVID-19 outbreak, telepsychiatry interventions have been widely considered for patients receiving routine psychiatric care [22,23], but few considered digital communication or video-conferencing solutions specifically for the purposes of providing mental health support for HCWs [24]. In the current pandemic context, HCWs presenting signs of depression, anxiety, distress, or PTSD need to be identified, with a view to offering them adapted therapeutic management as early as possible. In this context, psychiatrists of a French group of healthcare facilities offered a free teleconsultation service for staff (e.g., clinician-to-patient remote audio and video consultation), but the service was not widely used. The purposes of the teleconsultations could be for diagnosis, psychotherapy, and remote prescription. The low uptake of the service was in marked contrast with the overall impression of fatigue and exhaustion prevalent among the HCW teams. In this context, this study aimed to investigate the acceptability of telepsychiatry services as a means of providing mental health support to HCWs during the COVID-19 pandemic.

## 2. Materials and Methods

### 2.1. Design

In a multicentre, cross-sectional study, we used a 27-item questionnaire to investigate the acceptability to HCWs of a telepsychiatry service, as a means of providing mental health support during the COVID-19 pandemic. The survey was distributed online using the SurveyMonkey^®^ platform.

Based on a literature review of telepsychiatry, the questionnaire was developed by a panel of 8 experts including psychiatrists, geriatric medicine specialists, public health physicians, clinical research associates (CRAs), and one computer informatics researcher specialized in telemedicine. All were chosen for their expertise in matters relevant to the study, namely mental health management, telemedicine, methodology, statistics, and computing. The items investigated the respondents’ characteristics (7 items), the respondents’ attitudes towards technology (4 items), and telemedicine (2 items), particularly telepsychiatry (14 items). Respondents were asked about their representations of telepsychiatry, its impact on communication and physician-patient relations, and its perceived utility.

Responses were either yes/no questions, or on a 5-Point Likert scale: strongly disagree, disagree, neutral (neither agree nor disagree), agree, or strongly agree. The questionnaire is provided in Appendix A.

Our study was performed in 19 volunteer healthcare facilities (nursing homes, long-term care facilities, and psychiatric clinics) belonging to the same company. The staff of all these facilities were invited to participate by email, and information about the study was advertised in common areas using posters and flyers. For each facility, a local reference person was in charge of presenting the study to the staff and recruiting participants.

The study was performed in compliance with the Checklist of Reporting Results of Internet-E-Surveys (CHERRIES) [25].

### 2.2. Participants

All members of staff aged 18 years or over, and who started their employment at the healthcare facility prior to 1 March 2020 (the date on which access to healthcare facilities was restricted for the public, before being totally banned as of 11 March 2020) [26]) were invited to participate via a generic email. No other inclusion criteria were applied. The only exclusion criterion was refusal to consent to participation in the study.

### 2.3. Statistical Analysis

Quantitative variables are described as mean± standard deviation (SD) and categorical variables as number and percentage. Likert responses were dichotomized for easier interpretation, grouping “strongly disagree”, “disagree”, and “neutral” under the heading “disagree/neutral”, and grouping “agree” and “strongly agree” under the heading “agree”. Variables were compared using the chi square or Fisher’s exact test, as appropriate. Crude odds ratios were estimated with the Cochran-Mantel-Haenszel test. No assumptions could be made for the sample size calculation due to the lack of available literature data about the phenomenon under study. We aimed for 15 respondents per facility, in a purposive sample, for a target total of 285 respondents. A *p*-value < 0.05 was considered statistically significant. All analyses were performed using SPSS 21.0^®^ (SPSS, Chicago, IL, USA).

## 3. Results

Between October and December 2020, 321 responses were received out of 1233 (26% response rate), from the 19 participating facilities. All 19 facilities had at least 15 respondents except 3 (two had 14 and one 13), with a mean of 17 ± 2.5 respondents per facility.

Among the respondents, 278 (87%) were aged 25 to 60 years, 283 (88%) were females, and 201 (63%) were caregivers (physicians, nurses, and care assistants). Ninety-eight (31%) respondents declared that they had previously consulted a mental health professional (psychologist or psychiatrist) at least once in their life. The socio-demographic characteristics of the study population are shown in Table 1, and the full results are presented in Appendix B.

Overall, age was found to be significantly associated with the perception of teleconsultations (*p* = 0.037) and technological tools (*p* = 0.001), while female sex negatively impacted on the perception of teleconsultations (OR: 0.37 [95% confidence interval (CI): 0.17–0.84]). A total of 206 (64%) declared that they had patients infected with COVID-19 in their facility, while 208 (64%) declared that the current pandemic had an impact on their mental health.

A greater number of respondents reported negative mental health repercussions of the epidemic in facilities that received COVID-19 patients (OR: 1.96 [95%CI: 1.22–3.15]), especially when the respondents had previously consulted a mental health professional (OR: 2.58 [95%CI: 1.48–47.47]).

### 3.1. Perception of Technology

Almost all respondents (*n* = 314, 98%) had at least one device that would enable them to participate in a psychiatry teleconsultation (e.g., computer, tablet, or smartphone), an internet connection with sufficient capacity (*n* = 312, 97%), and a quiet room where they could spend at least 30 min in a teleconsultation (*n* = 292, 91%). Overall, 246 (77%) had already participated in a video conference (for personal or professional purposes), and 256 (80%) felt that they were sufficiently at ease with their devices to participate in a teleconsultation. In reality, very few participants (*n* = 67, 21%) had ever participated in a teleconsultation as a patient, while 191 (60%) considered a teleconsultation to be a secure procedure that would respect their privacy.

### 3.2. Perception of Telepsychiatry

A total of 244 respondents (76%) considered that telepsychiatry is useful to accompany people facing the COVID-19 crisis, but only 168 (52%) declared that they had easy access to appropriate psychological or psychiatric support. Only 48 respondents (15%) reported that they would be more at ease with tele-support than with presence-based support.

Regarding telepsychiatry, 195 (61%) answered that they would feel at ease talking about themselves, and 201 (63%) would feel at ease talking about their problems or receiving help for psychological issues.

However, 174 (54%) would not spontaneously discuss their preoccupations or would not accept the initiation of pharmacology therapy via teleconsultation. In total, 198 (62%) believed that a remote relation with a mental health professional would not suit them, while 144 (45%) believed that distance-based management would be identical to presence-based management.

The respondents who were least inclined to accept mental health support via telepsychiatry were caregiving staff (OR: 0.51 [95%CI: 0.28–0.94]), and those working in facilities receiving COVID-19 patients (OR: 0.59 [95%CI: 0.36–0.95]). These same groups were also least likely to spontaneously discuss their problems in a teleconsultation (OR: 0.54 [95%CI: 0.34–0.85]). Respondents who had previously consulted a mental health professional at least once in their lifetime would more easily accept initiation of medical therapy via teleconsultation (OR: 0.60 [95%CI: 0.38–0.95]).

## 4. Discussion

This study aimed to assess the acceptability of mental health support via telepsychiatry for HCWs in the context of the COVID-19 pandemic. Our results show that women, caregiving staff, and those directly involved in the care of COVID-19 patients are less favourable to the idea of receiving remote support. Without focusing on the remote nature of the support, previous studies have highlighted the same characteristics in people reluctant to receive support among HCWs [7,27,28]. Although more likely to suffer from psychological disorders, particularly due to chronic exposure to stress, HCWs are less inclined to seek psychological or psychiatric support [27]. Lai et al. reported that nurses, women, and frontline HCWs working in direct contact with COVID-19 patients had more severe degrees of mental health symptoms on all measures, compared to other HCWs during the epidemic [7]. Additionally, in the context of the COVID-19 pandemic, Chen et al. reported that despite clear signs of psychological disorders, medical staff in Wuhan (Hubei, China) at the centre of the epidemic were reluctant to participate in the group or individual psychology interventions that were offered to them [28].

To limit the impact of the COVID-19 pandemic on mental health, the American Psychiatric Association (APA) rapidly encouraged the wider use of telepsychiatry as a means to provide support [29]. In this context, Viswanathan et al. tested various individual and group care solutions among frontline clinicians [24] and reported good uptake among medical staff, who found the sessions helpful. In our study, despite the reported high acceptability of telepsychiatry among our participants, only 15% would be more at ease with remote support. To the best of our knowledge, ours is the first study to describe a discrepancy between the perceived utility of telepsychiatry as a means of providing support for HCWs, and the preferences the HCWs express for a presence-based consultation with a mental health professional.

A systematic review of telehealth interventions reported that the use of telepsychiatry in the general population appears to be feasible, acceptable, and effective compared to face-to-face consultations [30]. Our results suggest that the telepsychiatry solutions offered to HCWs should be adapted to take into account their specific relation to the question of mental health. In a study exploring the acceptability of new technologies in a sample of 515 psychiatrists, Bourla et al. described several barriers to the use of telemedicine solutions, including technological, logistic, clinical, and ethical issues [31]. Our results suggest that the barriers in our population were predominantly related to the clinical setting (e.g., only 46% would spontaneously address their preoccupations) or to ethics (as shown by only 60% agreeing that a teleconsultation is a safe form of interaction that respects their privacy). The preference for face-to-face consultations seems to reflect a fear that the physician–patient relationship will be perturbed by the use of new technologies [32,33]. To facilitate the creation of a productive patient–carer relationship using remote consultations, Smith et al. recommend in particular that the first consultation might best be performed face to face [34]. Finally, our results show that a prescription via a psychiatry teleconsultation would not be widely acceptable, with only 46% of HCWs reporting that they would accept a prescription of medical therapy after a teleconsultation. In contrast, prescriptions of antipsychotics and mood stabilisers remained at similar levels, despite a shift in the type of patient contact in the study by Patel et al., but the study population was composed of patients, and not HCWs as in our study [35]. Additionally, our study design precluded any assessment of whether respondents would more readily accept a face-to-face prescription.

Cultural factors may play a critical role in the diagnosis and management of mental illness [36], especially use of medications [37]. Culture is a set of understandings shared by a socially constructed group of people [36,38]. In our study context, there are at least three cultural biases that deserve to be considered, namely being French, being a HCW, and working in the same group of healthcare facilities. Cultural bias at the level of a country has been widely documented [39,40], and therefore, generalization of our results into other contexts should be done with caution. Although women are overrepresented in our study population, the population is nonetheless similar to the overall population of staff in nursing homes and long-term care facilities in France, since the latest census figures from 2015 reported 92% females among paramedical staff, 94% among nurses’ aides, and 59% among administrative staff, yielding an overall total of 87% women [41]. Conversely, in our study, all the respondents belonged to a single group of healthcare facilities, and therefore, had a standardized and similar work environment, which could induce cultural bias at an organizational level. Therefore, our results regarding telepsychiatry as a means for HCWs to talk about themselves, or their problems, or to spontaneously discuss their preoccupations, may be influenced by cultural factors, and further transcultural studies are needed to corroborate these results in other cultural settings.

Finally, it is known that representations about teleconsultation change after having tried it [42], yet few respondents in our study had ever previously participated in a remove consultation as a patient [21% (*n* = 67)]. Further studies are warranted among HCWs who have previously received remote support via telemedicine. If the acceptability of teleconsultation improves after trying it, perhaps the same might be valid for the acceptability of tele-prescription too, which should be balanced with the acceptability of face-to-face prescription.

## 5. Conclusions

This study aimed to assess the acceptability of mental health support via telepsychiatry for HCWs in the context of the COVID-19 pandemic. Our results show that caregiving staff and those directly involved in the care of COVID-19 patients are less favourable to the idea of receiving remote support. Females seem to be less favourable too, but this result should be interpreted with caution in view of the high proportion of women participating in the study. A number of obstacles to the use of teleconsultations as a means of providing support were identified, notably technological and logistical barriers. In the current pandemic context, where HCWs are particularly exposed to the risk of PTSD, telepsychiatry is a useful tool to consider for providing mental health support. Remote mental health support could be proposed systematically instead of, or in addition to face-to-face consultations, especially given that HCWs are almost all equipped with devices than would enable them to participate easily in remote mental health consultations.

## Figures and Tables

**Table 1 ijerph-18-10146-t001:** Characteristics of the respondents.

Characteristics		*n* = 321 (100%)
Sex	female	283 (88%)
Age	18–24 years	26 (8%)
	25–34 years	77 (24%)
	35–50 years	124 (39%)
	50–60 years	77 (24%)
	>60 years	17 (5%)
Type of facility	Nursing home	159 (50%)
	Long-term residential facility	82 (25%)
	Psychiatric clinic	80 (25%)
Profession	Caregivers	201 (63%)
	Other (administrative or technical staff, logistics, catering, etc.)	120 (37%)
My facility received patients infected with COVID-19	Yes	206 (64%)
I have previously consulted a mental health professional (at least once during lifetime)	Yes	98 (31%)
The COVID-19 crisis had an impact on my mental health	Yes	208 (65%)

## Data Availability

The data presented in this study are available on request from the corresponding author. The data are not accessible to the public due to French legislation which makes the investigators responsible for data processing.

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
