# Peer review of "Telepsychiatry to Provide Mental Health Support to Healthcare Professionals during the COVID-19 Crisis: A Cross-Sectional Survey among 321 Healthcare Professionals in France"

_ijerph, 2021, doi:10.3390/ijerph181910146_

Round 1

Reviewer 1 Report

This is a fairly well written paper about a reasonably done study. My comments for the authors' attention are below:

  1. In the method section, a sample size calculation and/or justification is needed.
  2. In the discussion section, culture has to be addressed, e.g., the possibility that French health care workers are different in this regard than other health care workers (and hence the need to study this issue transculturally).
  3. The English can be improved a bit, e.g., near the end of the paper the word more should be replaced by the word likely (immediately after the word more); there are a few more linguistic errors.

Author Response

Dear Reviewer 1, 
Thank you very much for your comments on our work. Please find enclosed in the Word document our answers to your comments. 

Kind regards, 

Reviewer 2 Report

General Comments: The article addresses an important issue considering the current pandemic context in which we live, concerns about the effects of the pandemic on the mental health of health care providers and studies a possible strategy for monitoring these professionals through telemedicine.

Introduction: The introduction is very succinct, although it is recognized that there may still be few studies on this matter, a more comprehensive idea of the current state of the art should be given.

The authors in the discussion refer to some studies.

Materials and Methods: Some information about the population and the sample should be given, as well as the inclusion and exclusion criteria of the participants.

Results: The presentation of the results could be clearer and more graphic, perhaps it would be appropriate to add a table with the inferential statistics and add to this table a simple/multiple regression model where it is possible to perceive the weight/behavior of each of the factors studied by the authors, possibly also using stratified analysis.

There is text information that cannot be confirmed in the table shown.

Discussion: The discussion should be reviewed, taking into account what was said in results.

Limitations seem more like strengths than limitations,

the authors should further explore this topic, as it seems to us that the study has other limitations in addition to those mentioned by the authors.

Conclusions: Reductive and questionable conclusion (88% of participants are women)

Author Response

Dear Reviewer 2,
Thank you very much for your comments on our work. Please find enclosed in the Word document our answers to your comments.

Kind regards,

Reviewer 3 Report

Thank you very much for an opportunity to review this timely and thoughtful piece on using telepsychiatry access mental health support during the Covid-19 crisis among health professioanls. The authors have presented novel and useful data, and have provided an in-depth discussion of the limitations (and opportunities) of telepsychiatry. My only query relates to the forms and the content of telepsychiatry consultations available to the healtcare workers during the pandemic. Can the authors provide more information about that?  

Author Response

Dear Reviewer 3,
Thank you very much for your comments on our work. Please find enclosed in the Word document our answers to your comments.

Kind regards,

Round 2

Reviewer 2 Report

The authors substantially improve the article, namely in the introduction, material and methods, and the conclusion discussion. Regarding the results, I still think that a simple/multiple regression model would benefit the article. 

Author Response

Revisions for submission : ijerph-1386215 “Telepsychiatry to provide mental health support to healthcare professionals during the Covid-19 crisis: a cross-sectional survey among 321 healthcare professionals in France”

Thank you very much for your comments. Below are our point-by-point responses to the remaining comments.

Kind regards,
